# Aerodynamic Study of a Drag Reduction System and Its Actuation System for a Formula Student Competition Car

**Ricardo Loução** [1], **Gonçalo O. Duarte** [1,2] **and Mário J. G. C. Mendes** [1,3,*]

1 ISEL—Instituto Superior de Engenharia de Lisboa, Instituto Politécnico de Lisboa, 1959-007 Lisbon, Portugal
2 Centre for Innovation, Technology and Policy Research (IN+), Associação Para o Desenvolvimento do Instituto Superior Técnico, Universidade de Lisboa, 1049-001 Lisbon, Portugal
3 Centre for Marine Technology and Ocean Engineering (CENTEC), Instituto Superior Técnico, Universidade de Lisboa, 1049-001 Lisbon, Portugal
* Correspondence: mario.mendes@isel.pt

**Abstract:** This work presents a computational fluid dynamic (CFD) analysis of a drag reduction system (DRS) used in a Formula Student competition vehicle, focusing on the interaction between the triple wing elements, as well as on the electrical actuators used to provide movement to the upper two flaps. The S1123 wing profile was chosen, and a 2D analysis of the wing profile was made. The trailing edge was rounded off to conform to Formula Student competition safety rules, resulting in around a 4% decrease in the lift coefficient and around a 12% increase in the drag coefficient for an angle of attack of 12°, compared to the original wing profile. The multi-element profile characteristics are: wing main plate with 4°, first flap 28°, and second flap 60°. To evaluate the wing operation, end plates and electrical linear actuators were added, generating a maximum lift coefficient of 1.160 and drag coefficient of 0.397, which provides around a 10% reduction in lift and a 9% increase in drag compared to the absence of the linear actuators. When activating the DRS, the flap rotation generates about a 78% decrease in the aerodynamic drag coefficient and 53% in the lift coefficient for the minimum aerodynamic drag setting.

**Keywords:** aerodynamic study; drag reduction system; wings actuators; formula student competition car

## 1. Introduction

The DRS system was introduced in Formula 1 in 2011, and it allows the rear wing flap to be placed at a horizontal position at certain locations on the track when the pilot is one second apart from the pilot in front. As a result, the aerodynamic drag force is reduced by about 20%, increasing the possibility of overtaking [1]. An 83% reduction in aerodynamic drag was achieved for a free-flow tested multi-element rear wing, while for a ground-effect tested multi-element front wing, it achieved a 70% reduction. Together, these two systems contributed to a 53% reduction in aerodynamic drag created by the car [2]. A similar application of DRS can be done to Formula student (FST) competition cars, leading to lap reduction times of circa 1.5 s tested on a track, covering 30 laps [3].

The wing profiles used in motorsport are identical to those used in aviation. Therefore, there are several different wing profiles for different types of use. Numerous aerodynamic studies can be found with different types of profiles aimed at motorsport, including a wide variety of NACA (National Advisory Committee for Aeronautics) profiles and others designed by Liebeck, Eppler Siegler, and Wortmann. However, and based on a broad research perspective, the application of some high lift wing profiles are very common, for example, the profiles, Eppler 423, S1223, FX74-CL5-140, are used due to their high lift coefficients in flows with a low Reynolds number [4].

During the last decades, some methodologies for creating high lift wing profiles were devised. In 1975, A.M. Smith [5] describes the physics of high lift wing profiles and how to achieve the best lift coefficient without separating its boundary layer, concluding that,

to obtain the highest lift coefficient without boundary layer separation, it is necessary to use a fast pressure recovery methodology, such as Stratford recovery or concave recovery. Unlike convex pressure recovery, concave pressure recovery reaches a very high adverse pressure gradient at the beginning of pressure recovery, causing the boundary layer to be in constant imminent separation [5]. However, because the boundary layer is so thin and sensitive, any disturbance causes the maximum lift coefficient to be greatly reduced. Another disadvantage is the sudden drop in lift once the profile exceeds the maximum angle of attack (AoA).

Numerical analysis of the profiles applied on FST vehicles can be done recurring to Computational Fluid Dynamics, which is a technique that can be used to simulate the behavior of fixed [6] or rotating [7] wings. Using numerical simulation and a criteria of low Reynolds number, high lift coefficient, and a good efficiency between the lift coefficient and aerodynamic drag, the best wing profile for FST application was found to be the S1123 profile due to its lift and aerodynamic resistance ratio [8]. Due to geometry restrictions implemented in Formula Student regulations (the trailing edge of the profile must have a minimum rounding radius of 1.5 mm, as sharp edges are not allowed for safety reasons), further improvements on the S1223 wing profile were made to implement it in a Formula Student race car [9]. The profile, S1223, was chosen after comparison with other profiles, due to its highest lift coefficient for Reynolds numbers between $2 \times 10^5$ and $3 \times 10^5$. The modified trailing edge of the S1123 profile was analyzed in 2D and 3D using Ansys® 2019 R2 software, Ansys Inc. As expected, the modified wing profile has lower aerodynamic coefficients ($C_L$ = 1.405; $C_D$ = 0.0510) [9] than the original S1123 profile ($C_L$ = 2.2; $C_D$ = 0.046) [10].

For the case of FST, it is important to have a multi-element system. The basic parameters to consider when designing a multi-element wing profile are the angles of attack and the gap between the main profile and the flaps. In order to achieve a highly efficient design, McBeath [11] indicates that "gap" and "overlap" values should stay between 1–4% and 1–6% of the profile chord size, respectively, which leads to the optimization of larger AoA. It is also suggested an angle of attack between 4 and 6% for the main wing, an angle of attack between 25 and 30% for the first flap, and 30–70% for the second flap [11].

The DRS operation requires an actuator, typically electric, to take advantage of the electrical systems of the FST vehicle. There are several types of actuators: linear or rotational, whose objective will be to transform their movement into the rotational movement of the two upper elements of the rear wing. However, the actuator shape, positioning, and components used to move the flaps also affect the aerodynamics of the rear wing. There was no information found on the impacts of introducing these elements on the aerodynamic characteristics of the rear wing. Consequently, this work aims to numerically simulate the structural and fluid dynamics, for the full functioning active DRS system (with a new linear electric actuator), using CFD tools. It was necessary to perform the following tasks:

1. Develop and model the multi-element profile, according to FST competition rules and regulations.

2. Develop a multi-element rear wing and study its aerodynamic behavior using finite element methods.

3. Develop a DRS System based on existing technologies in the literature that may be suitable for the FST competition.

4. Implement an active DRS System on the multi-element rear wing, using a new and different electric actuator (in a new position on the wing), and study its aerodynamic behavior using finite element methods.

This paper is defined by the following sections: in Section 1, a brief Introduction on the subject is done, considering the most used wing profiles and the information to build the multi-element wing, as well as the actuators used to the operation of a DRS system.

Section 2 presents the case studies, the actuator chosen, and the CFD domain characteristics. Section 3 provides the Results for the profile chosen, the multi-element profile,

and the 3D wing with and without actuator. Finally, Discussion analyzes the results and limitations of the study.

## 2. Materials and Methods

Figure 1 presents the main methodology regarding the numerical study of the rear wing with DRS actuators. Commercial software was used for geometry design (Solidworks® Student Edition 2020, Dassault Systemes) and to analyze the flow behavior (Ansys® 2019 R2, Ansys Inc.), following the flowchart presented.

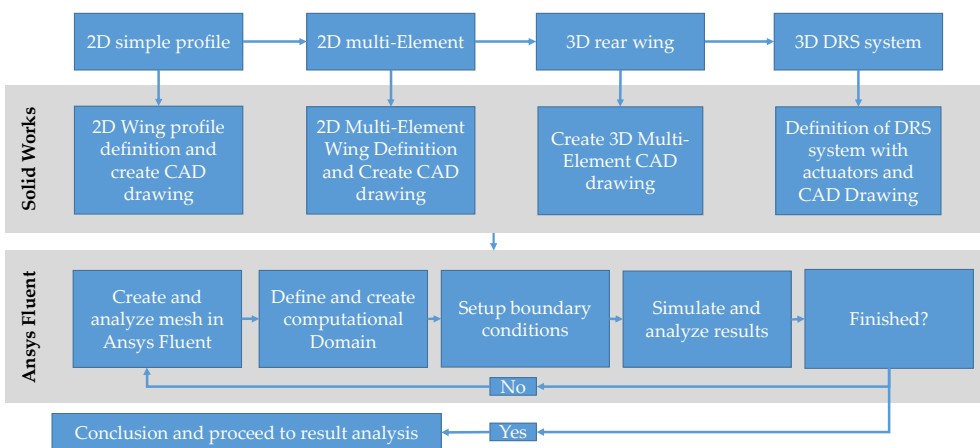

**Figure 1.** Flowchart with the geometries analyzed and software used.

### 2.1. Profile

Based on the literature review, the geometry of the rear wing will be designed with the profile S1223. In order to achieve an efficient design, following [11], a vertical gap of 3.8% and a horizontal gap of 5.2% of the profile chord size were used to create the multi-element profile. Additionally, angles of attack between 4° and 5° for the main element, between 28° and 32° for flap 1 and 60° and 65° there were studied for flap 2, also following [11].

The dimensions of the wings were defined, taking into account the regulatory restrictions of the Formula Student [12]. Thus, a chord length of 320 mm was defined for the main wing, with a 40% reduction for flap 1 and a 40% reduction for flap 2 compared to flap 1. The dimensions of the wing main, flap 1, and flap 2 can be seen in Figure 2a), while the dimensions of the horizontal and vertical gaps can be seen in Figure 2b).

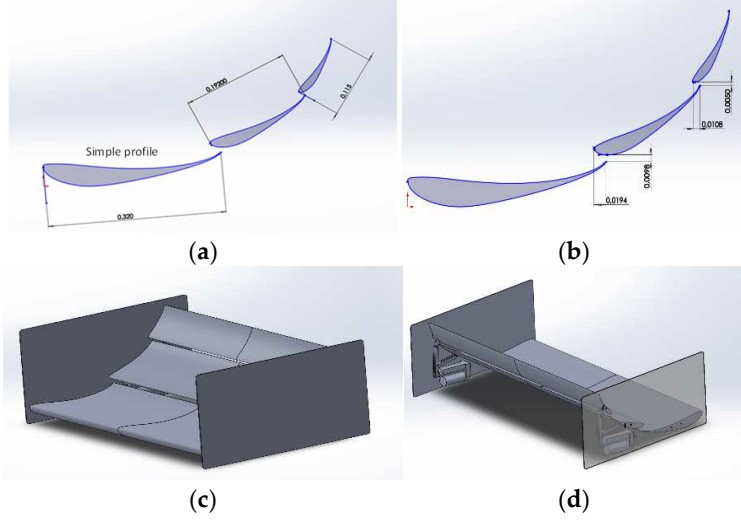

**Figure 2.** Case studies include: (**a**) simple profile; (**b**) multi-element profile; (**c**) 3D Wing without actuator; (**d**) 3D wing with two actuators.

There were four datasets performed, consisting of the S1123 profile (for validation), the S1123 multi-element profile (2D analysis, to define the best combination of AoA), the 3D wing without the actuators (Figure 2c), and with the actuators (Figure 2d). The full final rear wing consists of the wing profiles, endplates, and the DRS system actuators.

For each of the configurations studied, the lift and drag coefficients were calculated following Equations (1) and (2).

$$C_L = \frac{F_{Lift}}{\frac{1}{2}\rho U_\infty{}^2 A} \tag{1}$$

$$C_D = \frac{F_{Drag}}{\frac{1}{2}\rho U_\infty{}^2 A} \tag{2}$$

Equations (1) and (2) use information from air density ($\rho$), free flow air speed ($U_\infty$), and wing area ($A$). In these equations, it is used as the lift force for $C_L$ and the drag force for $C_D$. All the variables are in SI units.

### 2.2. Actuator

A LAS 1 electric linear actuator from TECNOPOWER was selected, with a stroke of 200 mm, weight of 1.04 kg, maximum propulsion and traction force of 1200 N, maximum holding force of 800 N, actuation speed between 8 and 12 mm/s, powered by 12 V and 6A, with protection against water class IP65 [13]. In this work, two actuators (placed symmetrically on the endplates) were used to rotate the wing with synchronism and to withstand the forces involved. Figure 3 shows the actuator and the respective mechanism responsible for transforming the linear movement of the actuator into a circular movement of the rear wing flaps. Rods 2, 3, and 4 are fixed between them and do not have individual freedom of movement. The system is closed when the actuator stem is at rest and the wings are at the maximum angle of attack, according to the design selected. When the linear actuator rod moves 6 mm, rods 2, 3, and 4 transform the linear movement into circular movement, rotating Flaps 1 and 2, having attack angles of 0°, and the system is considered open.

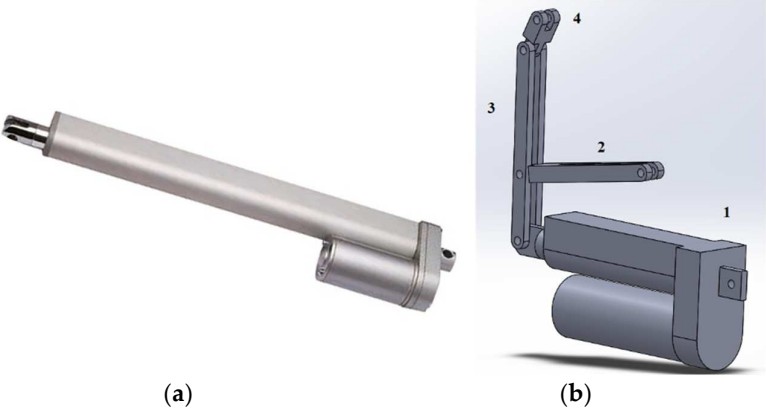

(**a**) (**b**)

**Figure 3.** (**a**) Actuator and (**b**) mechanism used to move the DRS system. Rods 2, 3, and 4 transform linear motion into circular motion.

### 2.3. CFD Domain

The configuration of the computational domain used in this study was obtained based on the best practices for automotive aerodynamic simulation, indicating that the domain must be at least three times the length of the car from the front and five times the length from the rear [14].

The S1123 multi-element profile 2D study domain has the following dimensions: 20 meters long and 10 meters high. In this case, as the aerodynamic analysis is only performed on the rear wing (isolated from the vehicle), the dimensions were defined through the minimum chassis length established in the Formula Student competition

regulations (1500 mm) [12]. Due to the complexity of the chosen wing profile, the virtual wind tunnel was designed with a length four times longer than the car from the front and eight times the length from the rear, which is very close to the minimum guidelines defined by Lanfrit [14]. The wing was centered in the middle of the virtual tunnel to increase the ability to capture the turbulent flow behavior caused by the wing.

The boundary conditions of the present study (Table 1) were defined to represent the track conditions in a Formula Student race. The fluid at the entrance of the virtual wind tunnel ("inlet") was defined with a velocity of 14.7 m/s with a direction normal to it. The fluid velocity is equivalent to a Reynolds number of $1 \times 10^6$, calculated for an S1223 wing profile with 1 meter of chord. The "walls" of the domain were defined as "no slip moving wall" to simulate free flow, with a translational velocity equal to the velocity of the fluid at the entrance of the domain.

**Table 1.** Boundary conditions.

| Type of Flow | 3D Steady State Flow |
|---|---|
| Turbulence model | Transition SST |
| Turbulence intensity | 0.003% |
| Turbulent Viscosity Ratio | 1% |
| Inlet velocity | 14.7 m/s |
| Wall Treatment | Automatic wall treatment |
| Wall | Stationary wall, Specified Shear |
| Rear Wing Wall | Stationary wall, No slip |

Atmospheric free flow simulations have very small turbulence intensities, often approximately 0.1%. Experimentally, it is verified that lower values do not modify the location of the laminar–turbulent transition. The turbulent viscosity ratio under boundary conditions is generally between 1 and 10 for external flows [15].

The turbulence properties at the entrance of the virtual wind tunnel were assigned again, according to the recommendations of Lanfrit [14], turbulence intensity of 0.003%, and a turbulent viscosity ratio in the order of 1. The virtual wind tunnel ("outlet") was configured to behave as a pressure outlet with a gage pressure of 0 Pa and with turbulence properties identical to the inlet. The "walls" of the domain were characterized as "Stationary Wall" and "Specified Shear" to simulate the rear wing in free mode.

The turbulence model chosen to develop the rear wing study was the Transition ($\gamma$-Re$_\theta$) SST model, due to its adequacy over other models used for the analysis of the behavior of the boundary layer for low Reynolds numbers [16], which is the case of Formula Student. This model has also been identified to provide more reliable results, clearly identifying the flow transition behavior for low Reynolds numbers, as well as for low and high angles of attack [17].

Equation (3) presents the transport equation for the transition momentum-thickness, where $\widetilde{Re}_{\theta t}$ is the transition onset momentum-thickness Reynolds number, $P_{\theta t}$ represents a source term that forces the $\widetilde{Re}_{\theta t}$ to match the local transition momentum-thickness Reynolds number, while $\rho$, $\mu$, and $U$ represent the density, dynamic viscosity, and flow speed, respectively [18].

$$\frac{\partial\left(\rho\widetilde{Re}_{\theta t}\right)}{\partial t} + \frac{\partial\left(\rho U_j \widetilde{Re}_{\theta t}\right)}{\partial x_j} = P_{\theta t} + \frac{\partial}{\partial x_j}\left[\sigma_{\theta t}\left(\mu + \frac{\mu_t}{\sigma_g}\right)\frac{\partial\widetilde{Re}_{\theta t}}{\partial x_j}\right] \tag{3}$$

In the present project, the mesh quality for each simulation was analyzed through its asymmetry and orthogonality [19], as well as the residuals. An unstructured mesh with tetrahedral elements were used in all domains studied, except for the prismatic elements within the inflation layers. Tetrahedral elements were used because they present the fastest way to generate a computational mesh with the least amount of memory required, which is an important feature when using state-of-the-art commercial PC configurations. Moreover,

due to the complex shape of the wing, with the flaps and DRS actuator, the unstructured mesh allows a better study of these areas. For the 3D studies, in order to increase the mesh density near the trailing edges, the area of worst asymmetry and mesh orthogonality, the surface elements associated with the trailing edges of the wing profiles, was dimensioned with 0.0008 m and a volumetric rate maximum growth rate of 1.1. Figure 4 presents a detail of the mesh for the 2D profile and 3D rear wing with the actuator.

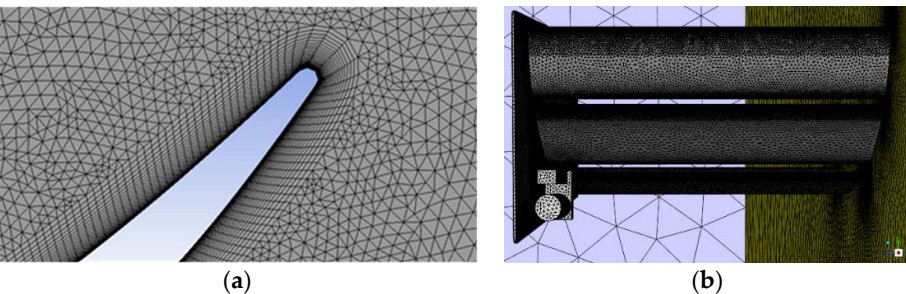

(a)　　　　　　　　　　　　　　　　　　　　　(b)

**Figure 4.** Example of the mesh detail: (**a**) trailing edge of 2D profile and (**b**) 3D rear wing with actuator.

### 3. Results

*3.1. S1123 Profile (2D)*

For the 2D simulation, an analysis of mesh properties was made, by changing the number of elements and evaluation of asymmetry and orthogonality, for three AoA (0°, 10°, and 12°). Figure 5 presents the impact of increasing the number of elements on $C_L$ and $C_D$ using the coarser mesh (112,908 elements) as reference.

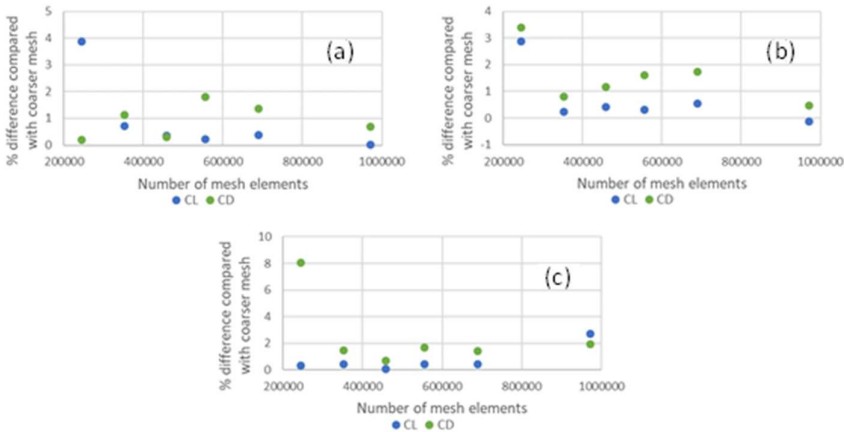

**Figure 5.** Analysis of mesh properties for three AoA: (**a**) 0°, (**b**) 10°, and (**c**) 12°.

Results suggest that increasing the number of elements above 352,358 elements provide small changes on $C_L$ and $C_D$ data. Therefore, this mesh was selected as the basis for the profile analysis since it uses low computational requirements and provides good quality on asymmetry (0.72573) and orthogonality (0.41782).

Comparing the results obtained through the aerodynamic simulations in the Ansys® 2019 R2 software, Ansys Inc. and the values generated in the XFOIL® v. 6.99 software, Massachusetts Institute of Technology (MIT), confirmed through the literature, deviations between 3.4% and 11.2% can be observed for the aerodynamic resistance coefficients and between 4.3% and 11.2% for the lift coefficients.

Figure 6a represents the comparison of the aerodynamic lift coefficient, while Figure 6b presents the comparison of the resistance coefficient, compared with the values obtained from the XFOIL® v. 6.99 software, Massachusetts Institute of Technology (MIT). In both cases, the similarities of the curves can be observed, such as a slight increase in the

aerodynamic drag coefficient and a small decrease in the lift coefficient. The differences found can be explained by the rounding of the trailing edge (according to the Formula Student rules) and following the literature, which has already stated that a small change in the profile can have a large impact on its aerodynamic behavior [20].

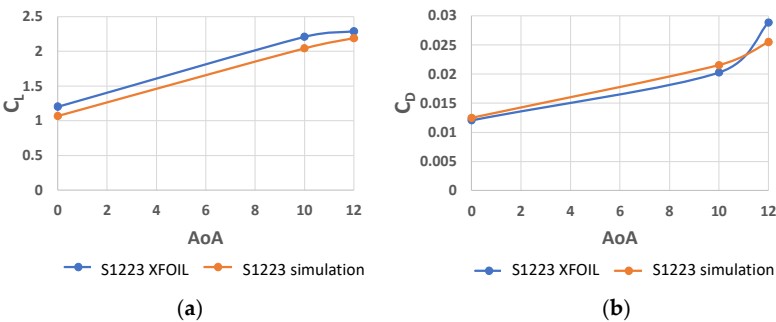

(a)                                                     (b)

**Figure 6.** Comparison of XFOIL data with simulated results: (**a**) $C_L$ and (**b**) $C_D$.

### 3.2. S1123 Multi-Element Profile

The best aerodynamic results obtained were $C_L$ = 3.4254 and a $C_D$ = 0.1223, resulting from a configuration (4°, 28°, 60°), that is, an AoA of 4° for the main wing, 28° for flap 1, and finally, 60° for flap 2. The aerodynamic results from this study can be observed and compared in Table 2. The mesh used had 481,875 elements and a good quality, with 0.8153 for asymmetry and 0.2930 for orthogonality.

**Table 2.** Aerodynamic study of the angles of attack of the multi-element profile.

| Angles of Attack (°) | | | Results | |
|---|---|---|---|---|
| **Mainplane** | **Flap 1** | **Flap 2** | $C_L$ | $C_D$ |
| 4 | 28 | 60 | 3.4254 | 0.1223 |
| 4 | 29 | 60 | 2.0829 | 0.2873 |
| 4 | 30 | 60 | 2.0545 | 0.2835 |
| 4 | 28 | 65 | 3.3851 | 0.1384 |
| 5 | 30 | 60 | 3.2454 | 0.1446 |
| 5 | 30 | 65 | 3.2489 | 0.1594 |
| 5 | 32 | 60 | 3.2637 | 0.1556 |

Observing the speed contour plot (Figure 7) around the multi-element profile generated by the Ansys® 2019 R2 software, Ansys Inc. after the simulation, the control of boundary layer separation is done by increasing speed due to the space between the elements near the low speed, prone to flow separation. The maximum speed obtained was on the mainplane of the multi-element profile with a velocity of 59.2 m/s, decreasing afterwards. Comparing with the maximum velocity of the profile S1223, studied in the previous section with an angle of attack of 12°, the maximum speed increased by 83.2%.

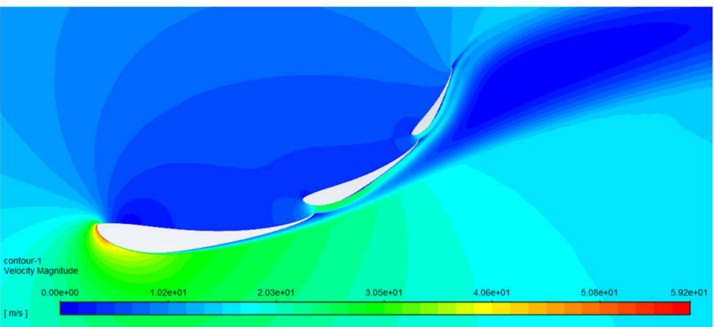

**Figure 7.** Multi-element profile speed contour graph.

### 3.3. S1123 Multi-Element 3D Wing without Actuator

The rear wing has a lift coefficient of 1.2806 and an aerodynamic drag coefficient of 0.4329. The flow behavior caused by the rear wing is expected, creating a low-speed zone represented in blue and a positive pressure zone, characterized by the orange color. Moreover, on the mainplane, a high-speed zone can be seen in red and yellow, as well as a negative pressure zone in light blue and green. Observing Figure 8, it can be seen that the endplates fulfill their role by preventing the appearance of large wingtip vortices on the edges of the wings and delaying their appearance, thus improving the ability to generate negative lift of the three elements belonging to the rear wing. However, vortices are formed on the upper and lower surface of the endplates with a vorticity that is 33% lower than those generated without the endplates.

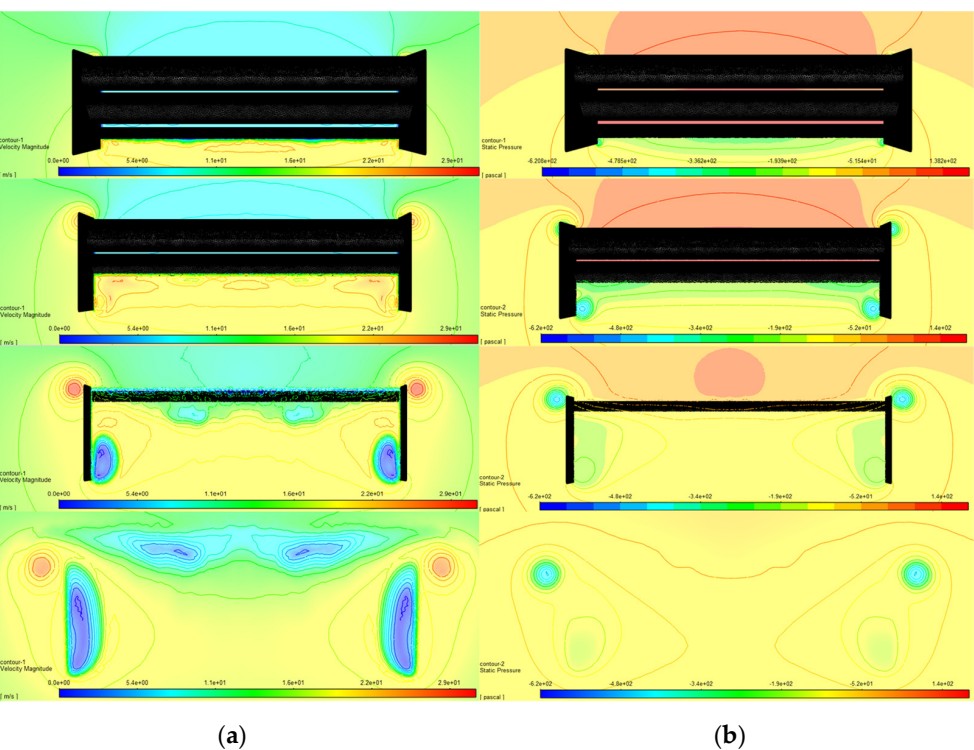

(**a**)　　　　　　　　　　　　　　　　　　(**b**)

**Figure 8.** (**a**) Speed contour and (**b**) pressure contour for the multi-element wing. From top to bottom, different sections of the rear wing are represented.

### 3.4. S1123 Multi-Element 3D Wing with Actuator

The actuator of the DRS system is responsible for a 9.5% decrease in the lift coefficient and 8.3% in the aerodynamic drag coefficient compared to the rear wing without an actuator. The rear wing plus DRS set has a maximum lift coefficient of 1.1597, while the maximum drag coefficient is 0.3969.

Figure 9 represents the speed and pressure generated on the rear wing with actuators by Ansys® 2019 R2 software, Ansys Inc. It can be observed that the linear actuator causes a deceleration of the flow around it, generating a very low speed and pressure zone in the immediate vicinity of the actuator. The vortex on the upper surface of the endplate has the same behavior, as presented in Figure 8. However, the vortex generated on the lower surface of the endplate has a larger core but circulates with a lower vorticity.

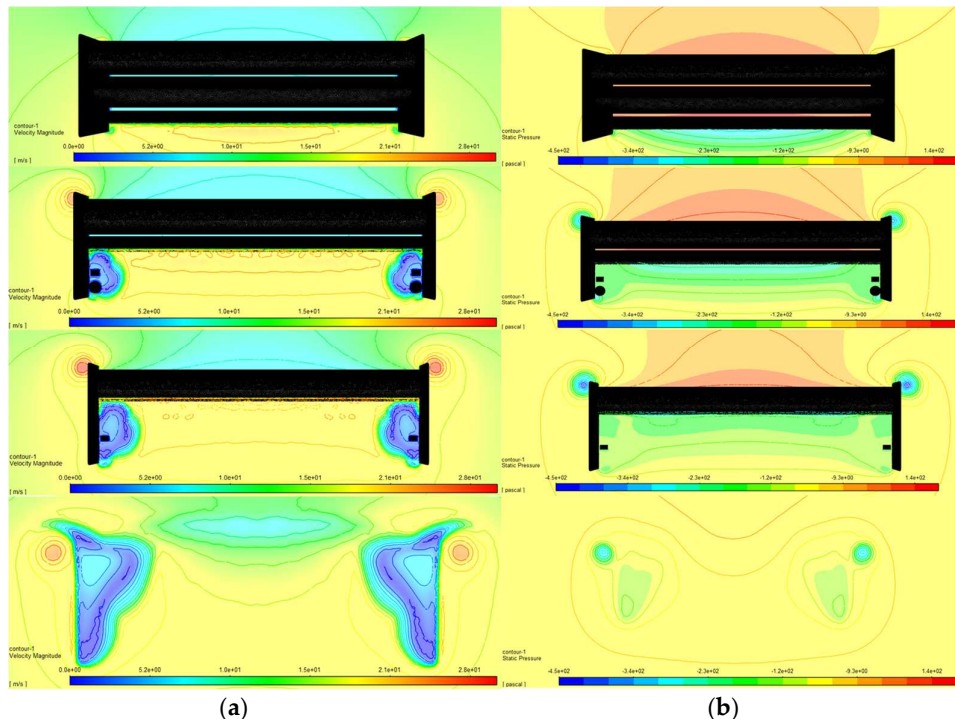

**Figure 9.** (**a**) Speed contour and (**b**) pressure contour for the multi-element wing with actuators. From top to bottom, different sections of the rear wing are represented.

When the DRS system is operated, the flaps rotate until they reach an angle of attack of zero degrees (Figure 10). When opening the flaps, the DRS system provides a 78% decrease in the aerodynamic drag coefficient, as well as a 53% decrease in the lift coefficient in relation to a closed DRS system. The rear wing, with the open DRS system, generates a lift coefficient of 0.5421 and an aerodynamic drag coefficient of 0.0864.

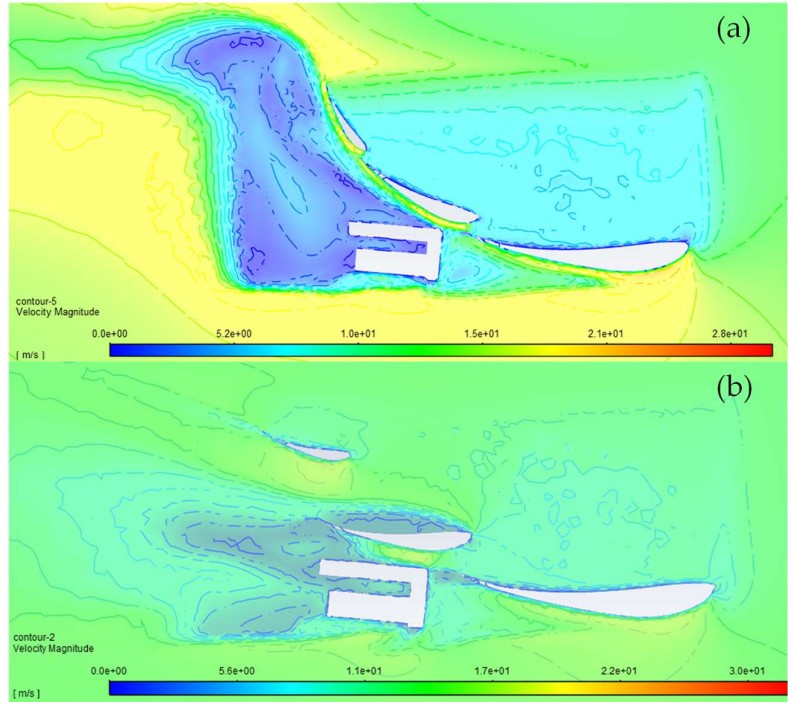

**Figure 10.** Speed contour near the endplate zone: (**a**) closed DRS; (**b**) open DRS.

Figure 11 presents the pathlines over the closed and open DRS system. When the DRS is closed it can be observed with a higher vorticity near the endplates, which is propagated behind the rear wing, increasing the drag. When the DRS is open, the flow is less affected by the vorticity near the endplates, particularly near the midsection of the rear wing. It is also suggested by the straight pathlines that the interaction between the two upper flaps is lower when the DRS is open, as expected.

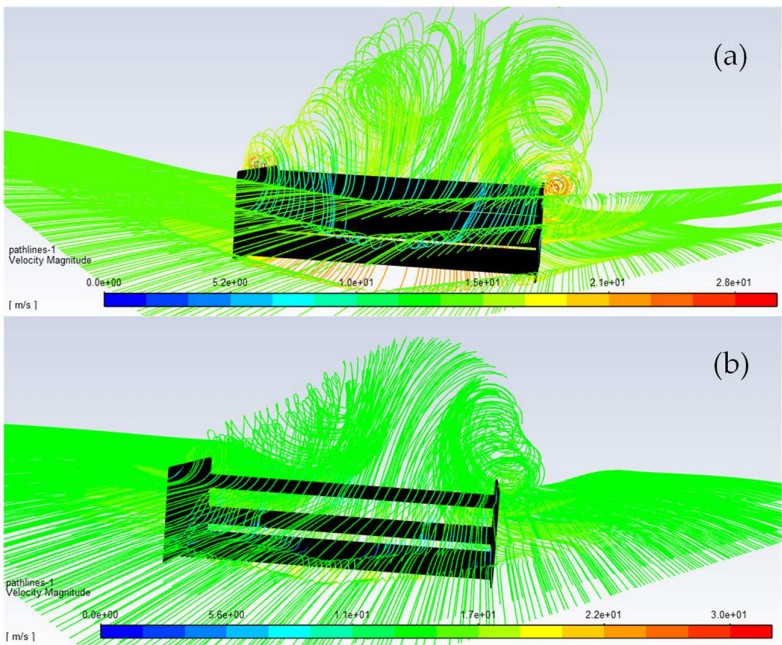

**Figure 11.** Pathlines: (**a**) closed DRS; (**b**) open DRS.

## 4. Discussion

This work presents the aerodynamic impacts of installing two electric actuators for DRS operation in a rear wing. It has been verified that rounding the edges of the S1123 profile to meet FST criteria generates a negative impact up to 11%, depending on the AoA.

It was also evaluated that the best configuration for the multi-element wing (4°, 28°, 60°, with $C_L$ = 3.4254 and $C_D$ = 0.1223) and the same configuration for the 3D wing ($C_L$ = 1.2806 and $C_D$ = 0.4329). It has also been shown the impacts of the physical implementation of the two actuators and the moving elements in the rear wing. This element is responsible for a 9.5% decrease in lift coefficient and an 8.3% increase in drag coefficient.

Numerical simulations of complex geometries use large computational resources. The quality of the data collected was verified using mesh asymmetry and orthogonality, as well as the residuals. For 2D simulations (single profile and multi-element) it was possible to obtain a mesh qualified as good [19] (below 0.8 for asymmetry and over 0.29 for orthogonality). The residuals are all below $10^{-4}$. Regarding 3D simulations, computational effort is much higher, limiting the asymmetry values to 0.96 and orthogonality to 0.004. These values generate a bad quality mesh, although residuals are below $10^{-4}$, excluding continuity, which was $10^{-3}$.

This paper also shows that CFD tools require computational efforts that, even for simple case studies, such as the one presented, need dedicated hardware to obtain better mesh layouts. Considering the context of a Formula Student study, the limitations presented indicate that CFD studies require several computing hours and hardware resources, limiting the final data quality. However, the results presented indicate that it is possible to evaluate the aerodynamic impact of the DRS system actuators and obtain the potential impacts of its introduction of the multi-element rear wing [21].

**Author Contributions:** Conceptualization, G.O.D. and M.J.G.C.M.; methodology, R.L, G.O.D. and M.J.G.C.M.; software, R.L.; validation, R.L., G.O.D. and M.J.G.C.M.; formal analysis, R.L, G.O.D. and M.J.G.C.M.; investigation, R.L.; resources, R.L.; data curation, R.L.; writing—original draft preparation, R.L, G.O.D. and M.J.G.C.M.; writing—review and editing, R.L., G.O.D. and M.J.G.C.M.; visualization, R.L.; supervision, G.O.D. and M.J.G.C.M.; project administration, G.O.D. and M.J.G.C.M. All authors have read and agreed to the published version of the manuscript.

**Funding:** This research received no external funding.

**Institutional Review Board Statement:** Not applicable.

**Informed Consent Statement:** Not applicable.

**Data Availability Statement:** The data presented in this study are available upon request to the authors.

**Conflicts of Interest:** The authors declare no conflict of interest.

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
