# Peer review of "Aerodynamic Study of a Drag Reduction System and Its Actuation System for a Formula Student Competition Car"

_fluids, doi:10.3390/fluids7090309_

Round 1

Reviewer 1 Report

Paper presents an Aerodynamic study of a drag reduction system for the formula car. Paper requires to address some of the following items:

I highly recommend authors to work on the quality of the images. The quality is so low that we are missing most of the physics presented.

Authors should dedicate a section for mesh study. Currently there is only one mesh presented and CFD results were carried out based on that mesh. How do you grantee that the mesh  is fine enough? or a coarser mesh might only sufficient to carry out the analysis?

What is the novelty of the work. What does it presents that others have not addressed? This should be clearly mentioned in the paper.

What are the boundary and initial conditions for the case. I highly recommend a table showing these information.

One might ask why SST model when you can use Reynolds Stress Turbulence Models? How SST compares to those turbulence models? a brief description requires by the authors to address this.

I honestly disagree with the comments about the CFD limitations described in the conclusion. There are many open-source software that you can leverage their capability. If you don't have a feature in Ansys software that doesn't mean it is hard. It is the software limitation not the CFD limitation.

I recommend manuscript publication after the major revisions I addressed above.

Reviewer 2 Report

In this manuscript, a computational fluid dynamic (CFD) analysis of a drag reduction system has been presented in a Formula Student competition vehicle. They focused on the interaction between the triple wing elements, as well as with the electrical actuators used to provide movement to the upper two flaps.

Suggestions that would improve the quality of the paper and changes which must be made before publication are as follows:

    Theoretical section:

a-1.  Why did authors apply unstructured mesh for modeling?

a-2. Please add mesh study for 2D & 3D simulations.

a-3. The authors should add new figures to validation of pressure and velocity distribution with published experimental data.

a-4. Which mothed do you apply for modeling the multi-element wing? Please explain about rotation specifications with two actuators.

a-5.  Please add a new figure for Velocity triangles at inlet and outlet of the multi-element wing.

a-6.  Which turbulence model did the authors apply? Explain with detail and new results.

a-7. Flowchart and algorithm of numerical solution should be added.

a-8. The introduction is not to the point. The study needs cite to some of improvements in CFD, such as (https://doi.org/10.1016/j.oceaneng.2019.106229).

    Format of the manuscript:

b-1. There are grammatical/typo errors in the manuscript.

Round 2

Reviewer 1 Report

Thank you for addressing my comments. Just a comment, open source software are not limited to personal workstation or paid cloud computing. They are eventually free and scale up linear.

Reviewer 2 Report

My comments have not applied in revised version.

Round 3

Reviewer 2 Report

All comments have been applied.